# An Overview of Cancer in Djibouti: Current Status, Therapeutic Approaches, and Promising Endeavors in Local Essential Oil Treatment

**DOI:** 10.3390/ph16111617

**Published:** 2023-11-16

**Authors:** Fatouma Mohamed Abdoul-Latif, Ayoub Ainane, Ibrahim Houmed Aboubaker, Jalludin Mohamed, Tarik Ainane

**Affiliations:** 1Medicinal Research Institute, Center for Studies and Research of Djibouti, IRM-CERD, Route de l’Aéroport, Haramous, Djibouti P.O. Box 486, Djibouti; mohamed.jalludin@gmail.com; 2Superior School of Technology of Khenifra (EST-Khenifra), University of Sultan Moulay Slimane, P.O. Box 170, Khenifra 54000, Morocco; a.ainane@usms.ma (A.A.); t.ainane@usms.ma (T.A.); 3Peltier Hospital of Djibouti, Djibouti City P.O. Box 2123, Djibouti; ibrahimhoumed@yahoo.fr

**Keywords:** cancer, Djibouti, essential oils, phytotherapy, status, treatment

## Abstract

Djibouti, a developing economy, grapples with significant socioeconomic obstacles and the prevalence of infectious pathologies, including certain forms of neoplasms. These challenges are exacerbated by limited access to affordable medical technologies for diagnosis, coupled with a lack of preventive interventions, particularly in disadvantaged areas. The attention devoted to local phytotherapeutic treatments underscores the uniqueness of Djibouti’s flora, resulting from its distinctive geographical position. International focus specifically centers on harnessing this potential as a valuable resource, emphasizing the phytoconstituents used to counter pathologies, notably carcinomas. This comprehensive overview covers a broad spectrum, commencing with an examination of the current state of knowledge, namely an in-depth investigation of oncological risk factors. Essential elements of control are subsequently studied, highlighting the fundamental prerequisites for effective management. The significance of dietary habits in cancer prevention and support is explored in depth, while traditional methods are examined, highlighting the cultural significance of indigenous essential oil therapies and encouraging further research based on the promising results.

## 1. Introduction

Cancer remains a significant global cause of mortality, and though strides have been taken, substantial headway in curbing its impact has proven elusive [1]. The genesis of cancers can be attributed to a trio of factors: erroneous dietary habits, genetic predisposition, and environmental influences. Remarkably, up to 95% of cancer occurrences stem from lifestyle choices, with a latency period of 20 to 30 years before symptoms manifest [2]. Insights from reputable sources such as the American Cancer Society and the International Union Against Cancer indicate that an estimated 12 million cancer diagnoses and 7 million deaths occur globally—a figure anticipated to double by 2030 (projected at 27 million cases with 17 million deaths) [3]. The challenge of efficaciously eradicating cancer and comprehensively managing its progression looms large. Earlier investigations have underscored the role of genetic anomalies, environmental factors, nutritional deficiencies, and metabolic stress as catalysts for most cancers [4,5].

In the realm of pharmaceutical interventions, the Food and Drug Administration (FDA) has granted approval to over 300 chemotherapeutic agents, including nivolumab, ipilimumab, pembrolizumab, 5-flurouracil, Olaparib, taxol, vinca alkaloids, and their derivatives, as well as gemcitabine and methotrexate [6,7,8]. Presently, the majority of chemotherapeutic treatments focus on single targets within nucleic acids, proteins, and carcinogenic signaling pathways. For instance, platinum drugs (such as oxaliplatin, carboplatin, and cisplatin) disrupt nucleotide synthesis and metabolism, inducing DNA damage [9]. Tyrosine kinase inhibitors like gefitinib, erlotinib, and icotinib target specific molecules. Angiogenesis inhibitors, such as bevacizumab, sunitinib, and sorafenib, impede the growth of new blood vessels. Yet, despite the development of an extensive array of drugs, cancer represents an ongoing evolutionary process [10,11,12].

The natural world has long served as an invaluable source for treating various cancer forms, with many remedies seamlessly integrated into daily diets [13]. These natural products confer substantial protection against diverse cancers and an array of other ailments. Medicinal plants rich in antioxidants safeguard cells from harm, yielding preventative benefits against cancer and other diseases [14,15,16]. Consequently, a diet abundant in antioxidant-rich fruits, vegetables, and herbs bears health-protective potential. Microbes and marine organisms, too, have emerged as significant contributors to cancer prevention and treatment [17].

Djibouti, owing to its strategic geographical location, boasts ecological diversity. With an enduring history of medical traditions and ancestral expertise in traditional herbal medicine, it stands as a prominent African nation. Herbal remedies have long been utilized to address ailments and infirmities in Djibouti, a practice that is still prevalent, particularly within rural areas [18].

This comprehensive overview focuses on recent advancements in relation to cancer in Djibouti, examining the current prevalence of the disease and its various manifestations. A thorough analysis of etiological factors reveals the complex interplay between environmental behaviors, genetic predisposition, and underlying molecular mechanisms. Furthermore, it delves into the fundamental pillars of oncological control, highlighting the importance of targeted interventions and meticulous monitoring to ensure optimal therapeutic outcomes. It also delves into conventional therapeutic modalities such as surgery, chemotherapy, and radiotherapy, highlighting their impact on tumor progression and overall patient survival. The critical importance of diets in modulating cellular signaling pathways associated with carcinogenesis is also highlighted, along with the examination of complementary therapies and phytochemical agents for their potential in managing side effects and improving the quality of life of cancer patients. At the same time, we highlight the cultural heritage of herbal treatments, providing in-depth insight into their biological relevance and potential contribution to the current therapeutic armamentarium. Finally, we underscore the promising results of in vitro studies on the use of compounds derived from Djibouti’s essential oils, elucidating their selective cytotoxic activity against specific cancer cell lines and their potential for innovative therapeutic interventions.

## 2. Materials and Methods

### 2.1. Collection of Information

This comprehensive overview comprises an examination of a diverse array of articles and books published up to the year 2020, sourced from prominent databases, including World Health Organization reports, Google Scholar, Scopus, PubMed, Science Direct, Wiley Online Library, and Springer. The research encompasses a meticulous search, employing keywords such as cancer treatments, medicine practitioner, dietary patterns, traditional medicine, traditional medicinal plants, and treatment of cancer by essential oils. These sources were carefully curated to provide a comprehensive foundation of information.

### 2.2. Medicinal Plants of Djibouti: Study Area and Identification

A comprehensive study involving phytochemical analysis and insights into medicinal utilization against cancer by local inhabitants across three regions of Djibouti (Figure 1) was conducted. This research focused on seven specific medicinal plants: *Cymbopogon schoenanthus* (L.) Spreng., *Lavandula coronopifolia* Poir., *Nepeta azurea* R.Br., *Ocimum americanum* L., *Ocimum basilicum* L., *Ruta chalepensis* subsp. *fumariifolia* (Boiss. & Heldr.) Nyman, and *Tagetes minuta* L. The investigation drew from pertinent articles within this field, serving as a foundation for the study’s exploration.

The process of identifying the botanical specimens was a collaborative effort undertaken by a dedicated team at the Medicinal Research Institute. This meticulous endeavor aimed to ensure accurate identification of the plant material, thus establishing the credibility and authenticity of subsequent findings.

### 2.3. Data Analysis

The process of data analysis employed a dual approach, involving both data extraction and data synthesis, facilitated through the utilization of narrative synthesis. This technique seamlessly integrated information from various sources, weaving together a coherent narrative. Additionally, tables played a pivotal role, concurrently summarizing essential information and study outcomes. These tables were skillfully crafted, aligning their headings with the core themes and objectives of the individual research studies. This methodological amalgamation not only ensured a comprehensive overview of the data but also allowed for a structured representation that aided in the effective comprehension of the synthesized information. Through this cohesive approach, the intricate web of data was meticulously organized, enhancing the accessibility and clarity of insights derived from the collected research.

In the specific context of the utilization of Djibouti’s essential oils in cancer treatments, Principal Component Analysis (PCA) serves as a mathematical tool aimed at unraveling intricate interdependencies among diverse parameters. The primary aim within this framework was to effectively visualize and consolidate data concerning the efficacy of IC50 values, as previously referenced, pertaining to Djibouti’s essential oils, along with vinblastine, doxorubicin, combrestatin A4, and monomethylauristatin E [19,20,21,22]. This endeavor was carried out to highlight divergences in values observed across the tested cancer cell lines, thereby facilitating a concise and comprehensible presentation. Specifically, PCA facilitated the categorization of the scrutinized cell lines into clusters, thus revealing associations through the active compound. In pursuit of this objective, coding methodologies were strategically employed [22], anchored in the *IC50* values extracted from each sample within the cancer cell lines. The intricate numerical dataset that underpins the foundation of this model was meticulously recorded and is detailed in Table 1.

## 3. Results and Discussion

### 3.1. Analysis of Cancer Incidence Statistics in Djibouti

Djibouti, a small country located on the northeast coast of the Horn of Africa, is undergoing rapid urbanization, with 80% of its population residing in urban areas, making it the most urbanized country in sub-Saharan Africa. Unfortunately, Djibouti faces economic challenges due to limited natural resources and geographic constraints that restrict agriculture, leading to high unemployment and budgetary difficulties. Additionally, recurring droughts severely impact the country [23].

A recent study lists the available medical equipment in African countries, including computed tomography (CT), magnetic resonance imaging (MRI), positron emission tomography (PET), gamma cameras, mammography, and radiotherapy equipment. Djibouti is classified among the countries with limited access to these essential medical facilities. Another study highlights the lack of access to radiotherapy in certain countries, including Djibouti, where demand for radiotherapy services is clearly identified.

Data on cancer care in Djibouti, though limited, were obtained from the national cancer profiles provided by the World Health Organization (WHO) (Table 2) [24]. According to the WHO report from 2020, Djibouti’s population is approximately 988,002 individuals, with 518,993 males and 469,009 females. During the most recent period, 765 new cancer cases were identified, with 286.479 cases among males and 479.286 among females. The age-standardized incidence rate per 100,000 population is 91.0, with a notable gender disparity: 71.5 for males and 112.9 for females. The overall risk of developing cancer before the age of 75 stands at 9.7%, with females facing a higher risk of 11.7% compared to males at 7.9%. Cancer-related deaths totaled 527, comprising 213 males and 314 females. The age-standardized mortality rate per 100,000 population is 65.3, with females at 76.9 and males at 55.2. The risk of dying from cancer before the age of 75 is 7.4%, reflecting an 8.5% risk for females and 6.4% for males [24]. Currently, there are 1328 prevalent cases of cancer, with 470.858 males and 857.470 females affected. The most frequent cancers, ranked by cases, include prostate cancer (males), colorectal cancer, non-Hodgkin lymphoma, leukemia, and liver cancer for both sexes. Among females, breast cancer, uterine cervical cancer, ovarian cancer, and colorectal cancer are prominent.

### 3.2. Cancer Risk Factors

Cancer, an intricate and multifaceted ailment, is influenced by a diverse array of factors, which collectively contribute to its development (Figure 2). These factors encompass various dimensions, including lifestyle choices, environmental surroundings, genetic predisposition, infectious agents, occupational exposure, and ionizing radiation. Recognizing the profound impact of these risk factors is pivotal for advancing comprehensive strategies aimed at diminishing vulnerabilities and heightening awareness to preclude the occurrence of cancer. A holistic comprehension of the manifold constituents contributing to the intricacies of cancer is imperative for crafting effective frameworks for prevention and control. In this endeavor, public health initiatives take center stage, prioritizing endeavors that encompass risk mitigation, early detection, and education. These initiatives play a pivotal role in ameliorating the escalating pervasiveness of cancer. By methodically addressing these underlying risk elements, we can catalyze significant progress in alleviating the toll that cancer exacts on individuals and communities across the globe [25].

The following sections delve into specific risk factors.

#### 3.2.1. Tobacco

The utilization of tobacco remains a widely acknowledged, substantial risk factor for a multitude of cancer types. The act of smoking introduces a cocktail of harmful substances into the body, including benzene, chromium, cadmium, acrylonitrile, 4-aminobiphenyl, 2-naphthylamine, and benzo[a]pyrene. These toxic compounds can inflict considerable damage on DNA, markedly heightening the probability of genetic mutations. These mutations, in turn, serve as a catalyst for the initiation and progression of cancerous cells [26,27,28].

#### 3.2.2. Obesity

The state of excess body weight is intimately linked to an augmented susceptibility to cancer development. Obesity can set off an elevated production and circulation of estrogen and insulin within the body. This hormonal surge provides an environment conducive to the unrestrained growth of cancer cells. This heightened risk is particularly conspicuous in the context of obesity-related cancers, such as those affecting the breast, colorectal, and endometrial regions [29,30].

#### 3.2.3. Air Pollution

The specter of air pollution has risen as a cause for concern, solidifying its role as an environmental risk factor for cancer. The constituents present in air pollutants, stemming from sources such as combustion processes and industrial emissions, have the potential to introduce carcinogens into the environment. Prolonged exposure to pollutants like radon and organic fibers has been closely associated with an elevated susceptibility to cancer [31,32,33].

#### 3.2.4. Genetics

Genetic factors wield considerable influence over an individual’s vulnerability to cancer. Discrepancies in genetic makeup, whether inherited or arising spontaneously, can substantially predispose an individual to certain cancer types. Families harboring a history of specific cancers may carry genetic mutations that substantially magnify their proclivity for developing the disease [34,35,36].

#### 3.2.5. Infectious Agents

Certain infectious agents have been firmly linked to an amplified risk of cancer. For instance, *Helicobacter pylori* (HP) is intricately associated with a nearly six-fold increase in the risk of developing stomach cancer. Similarly, the Human Papilloma Virus (HPV) stands as a primary catalyst for cervical cancer. These infections can precipitate chronic inflammation and engender genetic alterations that furnish fertile ground for the uncontrolled growth of cancerous cells [37,38].

#### 3.2.6. Occupational Exposure

Occupational exposure to noxious substances emerges as a pronounced concern in terms of cancer risk. Individuals immersed in work environments where they encounter carcinogens like diesel exhaust, asbestos, and polycyclic aromatic hydrocarbons are at heightened risk of cancer development. Factors such as passive smoking, solar radiation, and other workplace-related hazards further compound the overall incidence of cancer [39,40,41].

#### 3.2.7. Ionizing Radiation

Ionizing radiation, encompassing phenomena like X-rays, γ-rays, and neutron radiation, is well-recognized as a carcinogenic force. A myriad of sources—ranging from medical procedures to nuclear power production and atmospheric nuclear testing—can impart DNA damage and spur the inception of cancer. Even therapeutic applications, exemplified by high-dose radiotherapy for cancer treatment, introduce a potential for DNA damage and, consequently, the evolution of cancer through prolonged exposure [42,43,44].

### 3.3. Comprehensive Cancer Control: A Holistic Approach to Management and Care

Comprehensive cancer control strategies epitomize a systematic and integrated approach that combines prevention, early detection, treatment, palliative care, and survivorship endeavors. This orchestrated synergy is underpinned by the fusion of diverse components, encompassing the propagation of awareness regarding modifiable risk factors through public health campaigns, the establishment of streamlined screening programs to unveil the earliest signs of malignancy, the establishment of an equitable conduit to high-quality treatment, the cultivation of an environment that nurtures research and innovation, and the vigorous advocacy for policies that lend unwavering support to cancer prevention and care [45,46,47]. These interwoven elements coalesce into an encompassing framework that strives to surmount the multifaceted challenges posed by cancer, ultimately translating into optimized patient outcomes (Table 3).

At the crux of a resilient cancer care system is the bedrock of fundamental oncology services, orchestrating a harmonious process that revolves around patient-centered treatment. These indispensable services encompass the delicate art of precise diagnosis and staging, facilitated by the advanced arsenal of diagnostic tools, including cutting-edge imaging techniques and intricate molecular profiling [48]. Such diagnostic precision paves the way for the meticulous tailoring of treatment regimens, taking into consideration the intricacies of each patient’s condition. The quintessential multidisciplinary approach assumes prominence, wherein the collaborative efforts of medical, surgical, and radiation oncologists, coupled with the proficiency of supportive care specialists, collectively carve a path to comprehensive and individualized therapeutic strategies. Moreover, precision medicine emerges as a beacon of hope, steering treatment decisions through the genetic roadmap furnished by molecular profiling. This navigational prowess gives rise to therapies that delicately target the genetic anomalies fueling cancer progression, amplifying treatment efficacy while tempering the aftermath of side effects. The realm of palliative care, a pillar of paramount significance, extends to encompass pain and symptom management, creating a framework that improves patients’ quality of life. Simultaneously, psychosocial support services, including the provision of counseling and the haven of support groups, assist both patients and their families in navigating the turbulent emotional currents that accompany the dual specters of diagnosis and treatment [49,50].

In the luminous tapestry of cancer care, the recognition of early detection as an orchestrator of transformative change takes center stage. Through proactive screening initiatives and the amplification of awareness campaigns, the potential for redrawing the trajectory of treatment outcomes becomes vividly apparent. Timely identification, akin to a key that unlocks the doors to effective intervention, can metamorphose the very course of the disease. Yet, this transformative potential is only fully realized in a landscape where treatment access is not an aspiration but a guarantee. The labyrinthine corridors of cancer care must be devoid of barriers born from financial limitations or geographical divides, ensuring that each patient’s journey toward recovery is characterized promptness and quality of care rather than the constraints imposed by their circumstances. This very notion, of equitable treatment access, echoes the core tenets of fairness and inclusivity that are the linchpin of healthcare ethics [51,52,53].

Moreover, as patients traverse the intricate topography of cancer diagnosis and treatment, robust support mechanisms emerge as vital signposts. The comprehensive scaffolding of patient support, a mosaic of informational resources, financial aid programs, and emotional beacons, forms an unwavering lifeline. This support network serves as a guiding light, aiding patients and their families in navigating the often labyrinthine corridors of cancer care. In the realm of information, resources serve as guiding constellations, providing clarity amidst the bewildering terrain of medical terminologies and treatment options. Financial aid initiatives, a compassionate extension of care, alleviate the ancillary burdens that often accompany the diagnosis, ensuring that patients can focus on healing without the added weight of financial strife. Equally essential are emotional support mechanisms, whether in the form of counseling services or the embrace of support groups, which offer a safe haven where patients and their families can share experiences, find solace, and gather strength from collective resilience. This intricate network of support underscores the significance of an approach that places the patient at the epicenter, validating their emotional well-being as an integral facet of holistic care [54,55,56].

In the panoramic canvas of cancer care, the synthesis of these components unveils a narrative that goes beyond medical interventions. It encapsulates a philosophy that honors the dignity of each patient, addresses their multifaceted needs, and amplifies their prospects for recovery. The holistic and patient-centered approach, resonating across the spectrum from early detection to treatment access and patient support, casts a transformative light on the landscape of cancer care, ensuring that individuals facing the daunting terrain of cancer are met with unwavering support, empowered choices, and the assurance of comprehensive care (Table 4).

### 3.4. Medical Treatment of Cancer

Progress has been achieved in elucidating the intricacies of cancer pathogenesis, culminating in profound breakthroughs that hold significant promise for its therapeutic management. However, cancer’s tenacious and formidable nature persists, posing multifaceted challenges with intricate dimensions. These challenges encompass the profound inter-tumoral and intra-tumoral heterogeneity, as well as the complex orchestration of mutations across a myriad of genes, each contributing to the genesis and progression of malignancies. Furthermore, the pervasiveness of cancer across diverse cellular cohorts, ranging from epithelial to stromal and hematopoietic lineages, coupled with its propensity to infiltrate various organ systems, compounds the intricacy of this formidable disease landscape [57].

A pivotal consideration underscores the dynamic nature of cancer, characterized by an incessant evolutionary trajectory marked by the accrual of mutational load over time. Consequently, despite the exponential strides in knowledge, the clinical stewardship of malignancies in the 21st century remains ensnared in the web of escalating intricacy, rendering the treatment landscape a formidable domain to navigate. The armamentarium of conventional therapeutic modalities encompasses an array of approaches, including radiotherapy, surgical intervention, chemotherapy, proton therapy, and immunotherapy (Figure 3). Nonetheless, each of these avenues presents its unique constellation of limitations and prospective adverse ramifications [58,59].

Radiation therapy, a quintessential cornerstone of oncological intervention, entails the deployment of high-energy radiation to incapacitate neoplastic cells. Noteworthy in its prowess, this modality capitalizes on precision targeting within the confines of malignant tissue, thereby circumventing collateral destruction of adjacent healthy substrates. Nevertheless, prudence dictates a measured contemplation of its attendant downsides. The intricate geometries of energy dispersion might inadvertently entangle neighboring healthy cells, amplifying the potential for gratuitous injury. Furthermore, the indiscriminate scope of radiation deployment is concomitant with the specter of adverse effects, compromising patients’ well-being and convalescence [60].

Surgery, in turn, embodies the strategic excision of malignant tissue, operating under the overarching mandate of eradicating distinctly localized neoplasms. This approach, celebrated for its pinprick precision within the purview of affected regions, is not without its inherent demerits. Foremost among these is the proclivity for tumor cells to extend their invasive tendrils into neighboring tissues, thus heightening the specter of relapse. Furthermore, the landscape of outcomes is intrinsically contingent upon the geographic site of manifestation, often precipitating enduring disfigurement, engendering an indelible impact upon the patient’s appearance and quality of life [61].

Conversely, chemotherapy, an enthralling contender within the pantheon of cancer combat strategies, manifests through the administration of pharmacological agents with antineoplastic propensities. The paramount strength of this regimen lies in its temerarious potency in obliterating neoplastic entities, thus imposing a quelling restraint upon their unchecked proliferation. However, this therapeutic option is not bereft of concerning underpinnings. The lack of discrete target specificity elicits a cacophony of distinctive side effects, permeating the fabric of patients’ lives with a medley of afflictions. This array of grievances encompasses symptoms that include nausea, fatigue, alopecia, and other vicissitudes, endowing the therapeutic journey with a mosaic of tribulations for some afflicted individuals. While its merits against cancer cell annihilation are incontrovertible, the persistent pursuit of pinpointed drug distribution is the crux of oncological research, entailing the delicate balance of optimizing benefits and curtailing liabilities [62].

Proton therapy, an ascendant paradigm within the therapeutic arsenal, centers upon the directed application of proton beams, thus engendering a technologically refined variant of radiotherapy. A defining hallmark of this method resides in its adroit mitigation of noxious collateralities, coalescing with an enhanced spatial fidelity in targeting tumor foci, thereby exerting an amplified effect on treatment efficacy. Nonetheless, the landscape is marred by attendant caveats. Proton therapy proffers its benefits exclusively to a limited stratum of cancer subtypes, curtailing its transmutative potential. Additionally, the intricate machinery underpinning the establishment and operation of proton beam clinics materializes as a high-octane endeavor, imposing formidable fiscal requisites upon this therapeutic paradigm [63].

Immunotherapy, a paradigmatic example of biomedical innovation, operates within the theater of endogenous immune resources, marshaling the robust armamentarium of white blood cells and lymphoid system components to orchestrate a concerted attack against infectious incursions and pathogenic maladies. Embracing this strategy begets a tapestry of possible virtues. Chief among these is the tantalizing prospect of engendering an enduring reprieve from cancer’s scourge, contingent upon the amplification of immune efficacy. Moreover, the mien of this approach, underscored by a heightened propensity for precision, alludes to its conspicuous capacity to outshine its therapeutic counterparts in terms of localized effects. However, the herculean strides in the battle against cancer are shadowed by the specter of potential tribulations. Immunotherapy’s foray into the patient’s physiologic landscape presents the latent potential to catalyze severe or even fatal allergic responses, thereby underscoring the exigent imperative for judicious oversight and vigilant management throughout its clinical deployment [64].

In tandem, a constellation of emerging therapeutic paradigms has appeared on the horizon, buoyed by a promise that reverberates through the corridors of cancer therapy, attenuating suffering and slowing the grim march of cancer-related mortality. This constellation embraces the realms of photodynamic therapy, exploiting photosensitizing agents activated by precise wavelengths of light to dismantle tumor cells; photothermal therapy, orchestrated by photothermal agents harnessed to emit thermal energy upon light activation, engendering destruction of malignant entities; gene therapy, wielding genetic materials to stimulate the immune armamentarium or modify tumor cells themselves; and the incipient frontier of nanoparticle drug therapy, exemplifying the precision of targeted drug delivery mechanisms with the capacity to infiltrate the fabric of malignancies and render targeted cytotoxicity. These radical facets are expounded upon in subsequent elucidations, providing insight into the lexicon of evolving therapeutic frontiers in the enigmatic domain of oncological combat [65].

### 3.5. Drugs in Clinical Stages

Clinical cancer treatments encompass a diverse range of therapeutic strategies aimed at combating the disease at its molecular and cellular levels. Among the active agents used in these treatments (Figure 4 and Figure 5), chemotherapy remains a commonly employed approach, with agents such as methotrexate, doxorubicin, cyclophosphamide, paclitaxel, docetaxel, fluorouracil (5-FU), methoxetamine, gemcitabine, cisplatin, and carboplatin. Targeted therapies, on the other hand, focus on specific alterations present in cancer cells, utilizing drugs such as trastuzumab (Herceptin), rituximab (Rituxan), imatinib (Gleevec), erlotinib (Tarceva), and gefitinib (Iressa) [66].

Recent advances in oncology have also brought immunotherapies to the forefront, mobilizing the immune system to fight against cancer. Among these treatments are agents like pembrolizumab (Keytruda), nivolumab (Opdivo), ipilimumab (Yervoy), atezolizumab (Tecentriq), and durvalumab (Imfinzi). Concurrently, hormone therapy remains a key element in treating hormone-sensitive cancers, utilizing medications such as tamoxifen, anastrozole (Arimidex), letrozole (Femara), and leuprolide (Lupron) [67].

Protein kinase inhibitors, including vemurafenib (Zelboraf), sorafenib (Nexavar), and lapatinib (Tykerb), have gained importance in targeting specific signaling pathways within cancer cells. Monoclonal antibody-based therapies, including bevacizumab (Avastin), cetuximab (Erbitux), and panitumumab (Vectibix), are designed to disrupt crucial cellular interactions for tumor growth [68].

Moreover, other innovative therapeutic approaches have emerged, such as the use of bortezomib (Velcade) as a proteasome inhibitor and lenalidomide (Revlimid) as an immunomodulator, showcasing the ever-expanding array of strategies available to combat cancer. These diverse active agents reflect the constant evolution of oncology research and underscore the ongoing commitment to finding more effective and targeted treatments for cancer patients [69].

### 3.6. Complementary and Alternative Medicine and Dietary Patterns in Prevention and Supportive Cancer Care

In the ever-evolving landscape of cancer prevention and supportive care, alternative medicine (CAM) stands as an unwavering guardian of health, while complementary and dietary patterns emerge as a dynamic force, blending tradition and innovation [70].

#### 3.6.1. Complementary and Alternative Medicine (CAM): A Synergy of Tradition and Innovation

Complementary and Alternative Medicine (CAM) encompasses a diverse range of healthcare practices and products that fall outside the scope of standard treatments. CAM options available to cancer patients encompass acupuncture, aromatherapy, Ayurveda, cannabis, chelation, homeopathy, hypnotherapy, massage, naturopathy, traditional Chinese medicine, and other approaches and products. CAM is employed in conjunction with or in lieu of conventional therapies to enhance the well-being of individuals battling cancer. Over centuries, CAM has addressed a multitude of health challenges, including cancer. Despite controversies, patients consistently turn to CAM for a variety of reasons. Some integrate traditional treatments with alternative techniques to alleviate symptoms like headaches, nausea, fatigue, and anxiety. Others embrace CAM to counteract the adverse effects linked to certain cancer medications. Dissatisfaction with conventional treatments or a belief in the enduring solutions offered by CAM fuels this interest. Certain individuals harness CAM to bolster their immune system’s prowess in the fight against cancer. Ranging from acupuncture to dietary supplements, CAM encompasses an extensive array of treatments. While effectiveness varies, select CAM practices display potential in cancer care. However, a subset of CAM approaches may prove ineffective or even pose risks [71,72,73].

#### 3.6.2. Dietary Patterns: A Nutritional Tapestry for Health

Dietary patterns play a central role in cancer prevention and supportive care. Understanding the role of diets in preventing cancer and supporting treatments is essential for a better grasp of their impact on cancer-related health. This exploration illuminates foods rich in antioxidants, nutrients, and fiber, providing tailored dietary recommendations for cancer patients. The intricate interplay between dietary habits and cancer prevention becomes evident. Dietary patterns centered on consuming antioxidant-rich foods, such as colorful fruits and vegetables, nuts, and whole grains, have been scrutinized for their ability to counteract free radicals and reduce the risk of cellular mutations. Additionally, the nutrients present in these foods can bolster cellular repair mechanisms and tumor suppression. Integrating a balanced, fiber-rich diet is also crucial in maintaining a healthy gut microbiota, contributing to an optimal immune response against carcinogens [74].

For cancer patients, a tailored diet can play a pivotal supportive role throughout their treatment journey. Consuming protein-, calorie-, and nutrient-rich foods can counteract the weight loss and malnutrition often associated with aggressive treatments. Omega-3-rich foods, like fatty fish and flaxseeds, can contribute to alleviating the inflammation induced by cancer treatments [75]. Furthermore, diets rich in cruciferous vegetables, such as broccoli and cauliflower, are recognized for their capacity to induce detoxification and support the liver, playing a critical role in toxin elimination.

However, dietary recommendations for cancer patients must be personalized based on individual factors, including cancer type, nutritional status, and specific treatments. Side effects such as nausea, loss of appetite, and alterations in taste can influence dietary choices. In many instances, patients can benefit from tailored dietary guidance provided by healthcare professionals specializing in oncology [76].

### 3.7. Natural Productsfor Cancer Treatment

Over the past decade, a comprehensive compilation of studies has shed light on the ethnomedical and ethnopharmacological uses of a variety of natural species, whether of plant, animal, or mineral origin. These compounds, carrying significant medical implications, stand out in the field of oncology (Figure 6). Their diverse mechanisms of action have been supported by numerous experimental investigations, including in-depth biological assessments [77].

Particular attention is focused on arsenic oxide, which has been used for centuries in both Western and Chinese traditional medical practices. Starting in the 1970s, arsenic-based medications were employed to treat acute promyelocytic leukemia (APL) in China, marking the beginning of an era of arsenic treatments for this disease. Clinical data regarding the use of pure arsenic oxide were revealed in the 1990s. Subsequent clinical trials convincingly confirmed the notable advantages of arsenic oxide in patients with APL, thereby transforming this condition into a highly reversible disease. The combination of retinoic acid and arsenic oxide has profoundly revolutionized the fate of APL patients, achieving a remarkable cure rate of 90%. Further research has unveiled that arsenic oxide operates by degrading the PML-RARα fusion protein, responsible for oncogenesis in the context of APL, through a SUMO-dependent degradation pathway via the ubiquitination-proteasome system [78,79].

Other molecules derived from medicinal plants hold intriguing potential in the fight against cancer. Among these valuable discoveries, emodin extracted from *Aloe vera* (L.) Burm.f., artemisinin and artesunate from *Artemisia annua* L., and betulinic acid and betulin from *Betula* spp. stand out as promising agents. Similarly, berberine from *Berberis vulgaris* L., epicatechins and epigallocatechin from *Camellia sinensis* (L.) Kuntze, crocetin, crocin, and safranal from *Crocus sativus* L., ingenolmebutate from *Euphorbia peplus* L., geniposide and genipine from *Gardenia jasminoides* J.Ellis, apigenin and chamomillol from *Matricaria chamomilla* L., and panaxadiol and ginsenosides from *Panax ginseng* C.A.Mey., have all been meticulously examined for their anticancer properties. Additionally, compounds such as cryptotanshinone, salvianolic acid, and salvicine from *Salvia prionitis* Hance, silibinin and silymarin from *Silybum marianum* (L.) Gaertn., and paradol and shogaol from *Zingiber officinale* Roscoe further enrich this captivating array. These natural treasures have garnered sustained interest due to their promising mechanisms for targeting and disrupting fundamental cell growth and proliferation processes [80,81,82].

Taxanes, including paclitaxel and docetaxel, originate from the *Taxus* spp. tree bark and function as cellular regulators, stabilizing microtubules, slowing tubulin depolymerization, disrupting the cell cycle at G2/M, and inducing cell death; marketed as Taxol^®^, Taxotere^®^, Abraxane^®^, Jevtana^®^, Taxoprexin^®^, and Xytotax^®^, they span initial treatments to Phase I–III trials, addressing various cancers like breast, ovarian, and Kaposi’s sarcoma. Extracted from *Catharanthus roseus* (L.) G.Don leaves, Vinca alkaloids are semi-synthetically produced, operating with a dual mechanism: inhibiting tubulin polymerization while dismantling the mitotic spindle.Marketed as Vinorelbine, Vincristine, Vinblastine, Vindesine, Vinflunine, Vincamine, and Vintafolide, they are under clinical evaluation and are combined in trials for increased effectiveness. Similarly, Camptothecin and Irinotecan, from *Camptotheca acuminata* Decne. leaves, bind to the TOP1 cleavage complex, leading to DNA breaks and apoptosis during the S phase; they are used against ovarian, lung, and colon cancers as Topotecan, Irinotecan, and Belotecan in frontline therapy. Analogues of Podophyllotoxin, from *Podophyllum* spp. roots and rhizomes, block metaphase cell division, contributing valuably to lymphoma and testicular cancer clinical research. In addition, Roscovitine, extracted from *Raphanus sativus* L., inhibits cyclin-dependent kinases, slowing the cell cycle; Phase II trials are being conducted in Europe for Roscovitine and Seliciclib [83,84,85,86].

Another category of botanical bioagents plays a significant role, highlighting targeted activity against cancer checkpoints. For instance, analogues of hematoxylin, derived from the heartwood of *Haematoxylon campechianum* L., have emerged as competitive ATP inhibitors of broad-spectrum tyrosine kinases; they are notable for their potency, with IC50 values in the nanomolar range. Eucalyptin A, derived from the fruits of *Eucalyptus globulus* Labill., a widely distributed plant in southwestern China, has demonstrated marked inhibitory action on the HGF/c-Met axis. Pseudolaric acid B, a diterpenoid isolated from the bark of the root of the *Rhododendron kaempferi* var. *kaempferi* tree (Pinaceae), exhibits anti-angiogenic activity through a mechanism involving coordination between hypoxia-inducible factor 1-alpha (HIF-1α) and c-Jun. Parthenolide, a sesquiterpene lactone originally isolated from feverfew (*Tanacetum parthenium* (L.) Sch.Bip.), has revealed its potential for inhibiting the Wnt/β-catenin signaling attributed to its action on the ribosomal protein RPL10. Moreover, compounds derived from *Euphorbia peplus* L. have been discovered for their ability to modulate lysosome biogenesis [87,88,89,90].

Simultaneously, natural marine products, particularly those derived from marine invertebrates, are emerging as a possible source of anticancer drugs. Trabectedin, derived from the Caribbean tunicate *Ecteinascidia turbinate*, represents the first marine-derived anticancer drug, although its mode of action remains partially enigmatic. Researchers have discovered several new dimeric trabectedin alkaloids and their monomers in marine nudibranchs and their sponge prey. In-depth analysis of these compounds has deciphered the mechanism of NF-κB inhibition, thus shedding light on a better understanding of trabectedin. Additionally, marine polycyclic compounds could also pave promising paths. The synthesis focused on the function of the polyketide ulocladol, isolated from fungi associated with *Ulocladium botrytis* sponges, has highlighted a class of inhibitors of the M2 isoform of pyruvate kinase (PKM2), a crucial metabolic enzyme in the context of cancer [91,92,93,94].

### 3.8. Use of Essential Oils from Djibouti in Cancer Treatments (In Vitro)

Cancer remains one of the major public health challenges on a national scale in Djibouti. Despite significant advancements in clinical treatments, it remains crucial to explore new therapeutic approaches. With this perspective in mind, our study focused on evaluating the therapeutic potential of seven essential oils extracted from local medicinal plants in Djibouti as potential treatments against cancer. To provide context to our research, we compared these extracts to active compounds commonly used in cancer treatment protocols [19,20,21,22].

To conduct our investigations, we performed in vitro experiments. Initially, we carefully selected seven specific medicinal plants from Djibouti: *Cymbopogon schoenanthus* (L.) Spreng., *Lavandula coronopifolia* Poir., *Nepeta azurea* R.Br., *Ocimum americanum* L., *Ocimum basilicum* L., *Ruta chalepensis* subsp. *fumariifolia* (Boiss. & Heldr.) Nyman, and *Tagetes minuta* L. This selection was based on in-depth ethnobotanical surveys conducted by the team at the Institute of Medicinal Research of the Center for Studies and Research in Djibouti (Table 5). Subsequently, we extracted the essential oils using rigorously standardized methods and subjected them to tests on 13 distinct cancer cell lines: A2780, A549, HCT116, HEK293, JIMT-T1, K562, MIA-Paca2, MRC-5, NCI-N87, PC3, RT4, U2OS, and U87-MG (Table 6).

The choice of each cell line was guided by its relevance to identified national health issues. The A2780 cell line, derived from human ovarian carcinoma, is frequently used in ovarian cancer research and associated treatments. A549 cells, originating from human lung adenocarcinoma, are used for the study of lung diseases and viral responses. As for the HCT116 cells, they offer a model for exploring colorectal cancer, signaling pathways, and genetic regulation. The HEK293 cell line, derived from human embryonic kidney cells, proves valuable in molecular biology, biotechnology, and recombinant protein production. JIMT-T1 cells, from human mammary adenocarcinoma, are primarily employed in breast cancer research and targeted therapies. K562 cells, from human chronic myeloid leukemia, are frequently used to study leukemia, cell differentiation, and stem cells. MIA-Paca2 cells, derived from human pancreatic adenocarcinoma, play a crucial role in pancreatic cancer research and therapeutic approaches. The MRC-5 cell line, composed of human diploid lung fibroblasts, is used in virological research and vaccine production. NCI-N87 cells, from human gastric adenocarcinoma, are mainly exploited for gastric cancer research and the associated molecular mechanisms. PC3 cells, derived from human prostate adenocarcinoma, are commonly used to study prostate cancer and metastasis processes. RT4 cells, from human bladder carcinoma, serve as a model for bladder cancer research and therapeutic approaches. The U2OS cell line, derived from human osteosarcoma, is frequently employed to study cell biology, with a focus on differentiation and genetic regulation. Finally, U87-MG cells, derived from human glioblastoma multiforme, are extensively used in brain tumor research and neurobiology. Each cell line possesses specific characteristics that make it valuable tool for specific research domains, thus providing crucial insights for understanding disease and potential therapy development.

The cytotoxicity data from all the studies conducted in this regard, presented in the form of inhibitory concentration values (IC50), indicate the dose required to reduce cellular viability by 50% and provide insight into the relative effectiveness of the tested compounds, essential oils, and active principles on different cell lines. These summarized results are presented in Table 7.

Certain cell lines demonstrated notable sensitivity to the studied essential oils. For instance, the essential oils from *Cymbopogon schoenanthus* (L.) *Spreng*., exhibited strong cytotoxic activity against the A2780, HEK293, and U2OS cell lines, with notably low IC50 values. This suggests that these essential oils might be particularly effective against these specific types of cancer. On the other hand, the essential oils from *Lavandula coronopifolia* Poir. showed significant activity against the A549, MRC-5, and HEK293 cell lines. These findings suggest potential use in the treatment of these types of cancer while preserving the viability of non-cancerous MRC-5 cells. The essential oils from *Tagetes minuta* L. also demonstrated notable cytotoxic activity, especially against the K562, RT4, and U87-MG cell lines, with relatively low IC50 values. These observations indicate that *Tagetes minuta* L. could be a promising avenue for the treatment of certain cancers. However, it is important to note that some cell lines exhibited relative resistance to the tested essential oils. For example, the HCT116 and JIMT-T1 cell lines showed higher IC50 values for most essential oils, suggesting reduced reactivity to these specific compounds.

The modeling of IC50 values was conducted through principal component analysis (PCA), based on the following approximations: cytotoxic activity tests are independent, and effectiveness was coded based on the obtained IC50 values. Table 8 presents the new numerical coding parameters for the effectiveness of the tested products against cancer cell lines. Thus, the PCA correlation among the tested products is depicted in Figure 7 along two axes, F1 and F2, with retained variabilities of 29.87% and 19.38%, respectively, corresponding to a total of 49.25% of the information. Correlations between the tested products and cancer cell lines are presented in Figure 8. Furthermore, Figure 9 illustrates the Hierarchical Ascendant Classification (HAC) of the studied essential oils based on effectiveness.

Based on the analyses of the numerical values for cancer treatment, it is evident that the seven medicinal plants along with the four active principles can be grouped into three major clusters:Group 1: Contains the three active principles, namely Combrestatin A4, Doxorubicin, MMAE, and Vinblastine, used in cancer treatment, indicating the efficacy of this group in yielding positive cancer outcomes.Group 2: Encompasses the six essential oils *Cymbopogon schoenanthus* (L.) Spreng., *Lavandula coronopifolia* Poir., *Nepeta azurea* R.Br., *Ocimum americanum* L., *Ocimum basilicum* L. and *Tagetes minuta* L., suggesting promising results across all cell lines.Group 3: Comprises the MMAE active principle and the essential oil from *Ruta chalepensis* subsp. *fumariifolia* (Boiss. & Heldr.) Nyman, indicating a similarity in activity against all cell lines tested.

### 3.9. Limitations of the Current Knowledge in Djibouti—Perspectives

The current limitations in the understanding of cancer in Djibouti encompass several critical aspects, reflecting a deficiency in comprehensive data regarding the precise incidence and prevalence of diverse cancer types in the region. Furthermore, there exist substantial gaps in comprehending the specific contextualized risk factors, which hinders the development of targeted preventive strategies. Concurrently, the accessibility to adequate healthcare services and effective treatments for cancer patients is impeded due to numerous obstacles, contributing to a notable disparity in the quality of care. Additionally, the constrained availability of financial, technological, and human resources poses a significant impediment to conducting extensive research endeavors and implementing robust screening and prevention programs. This resource scarcity restricts the capacity for robust epidemiological studies and the exploration of advanced diagnostic methodologies, hindering the advancement of cancer care in the region. Moreover, the limited knowledge regarding traditional medical practices and the potential utilization of indigenous remedies within the framework of cancer management poses an additional challenge. Integrating traditional medicine with contemporary oncological approaches remains constrained due to the scarcity of validated scientific evidence, impeding the development of comprehensive and holistic treatment regimens. To address these limitations, a concerted effort to augment data collection, enhance research infrastructure, and foster collaborative partnerships is imperative, thereby facilitating a more comprehensive understanding of the multifaceted dynamics of cancer in Djibouti and paving the way for tailored interventions and improved patient outcomes.

## 4. Conclusions

The scientific study at hand provides a comprehensive examination of various facets concerning cancer and its treatments, with a specific focus on the context of Djibouti. It delves into a thorough analysis of cancer incidence statistics within the region, shedding light on pressing public health concerns. Identifying cancer risk factors, the study offers insights into contributing elements such as lifestyle, dietary patterns, and environmental influences. Moreover, it underscores the significance of adopting a holistic approach in cancer management, emphasizing the need for a comprehensive control strategy. The investigation delves into medical treatments for cancer, including drugs in clinical stages that reflect advancements in the pursuit of more effective therapies. Complementary and alternative medicine, alongside dietary patterns, are addressed as components of preventative and supportive cancer care. The study also highlights the intriguing potential of natural products, particularly essential oils, in cancer treatment through in vitro studies. This work contributes to a nuanced understanding of cancer dynamics in Djibouti, covering a spectrum from risk factors to the exploration of local essential oils for potential therapeutic benefits. It underscores the importance of continuous research and the integration of natural resources in medical approaches to cancer care.

## Figures and Tables

**Figure 1 pharmaceuticals-16-01617-f001:**
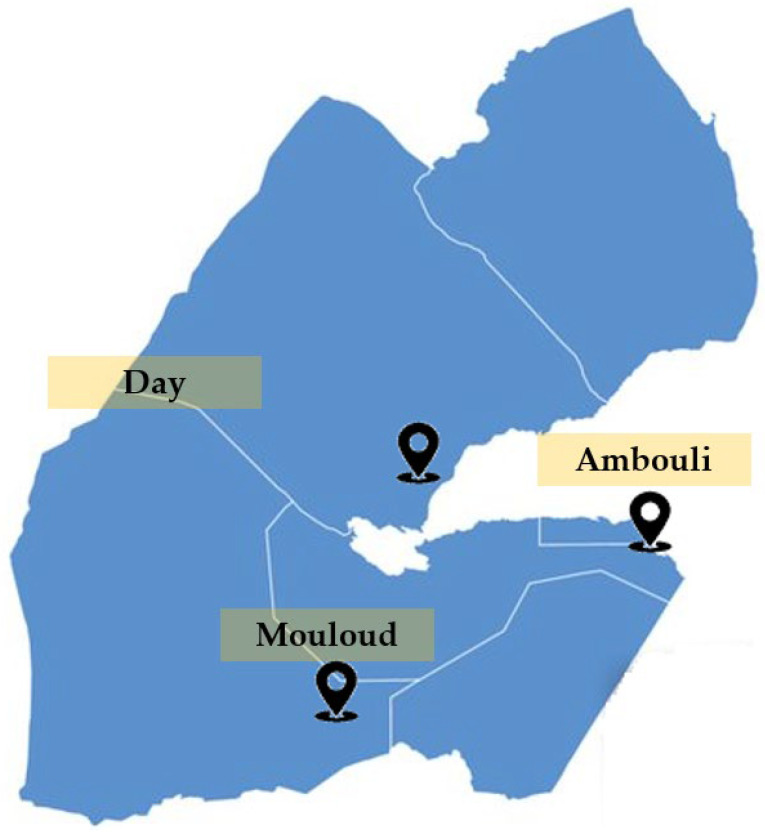
Place of collection of plants used for anticancer evaluation.

**Figure 2 pharmaceuticals-16-01617-f002:**
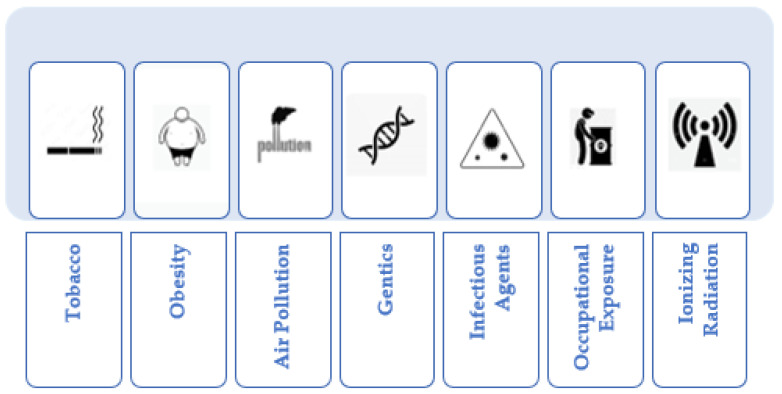
Principal cancer risk factors.

**Figure 3 pharmaceuticals-16-01617-f003:**
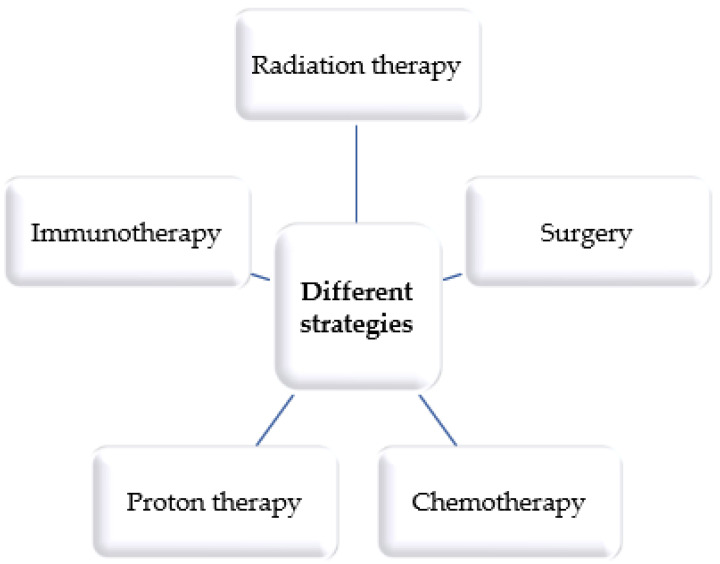
Main cancer therapies.

**Figure 4 pharmaceuticals-16-01617-f004:**
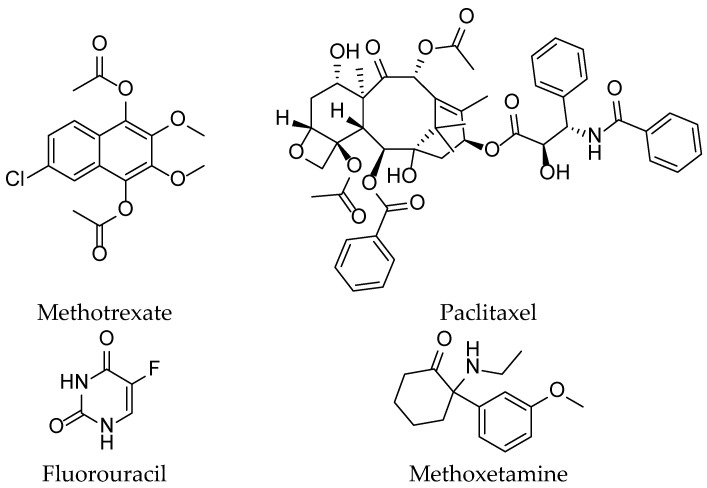
Some chemical structures of the active ingredients of drugs used in clinical cancer therapies.

**Figure 5 pharmaceuticals-16-01617-f005:**
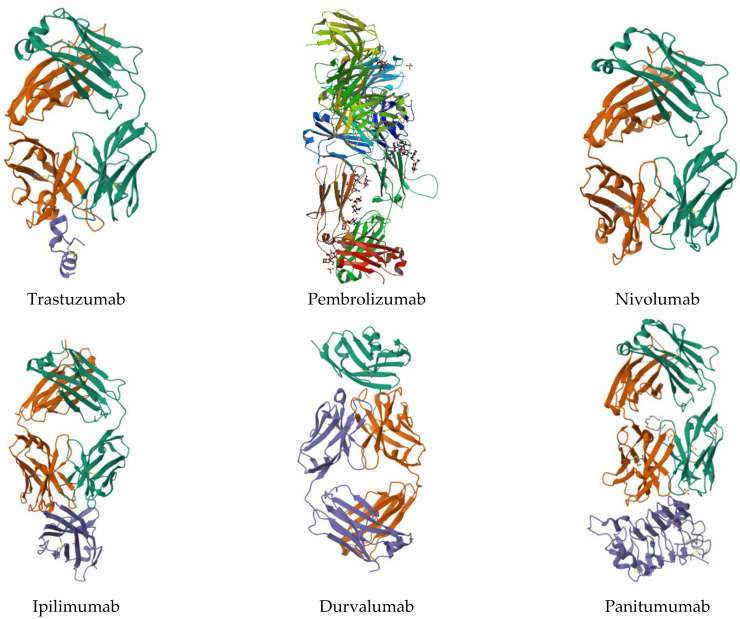
Some protein structures used in clinical cancer therapies.

**Figure 6 pharmaceuticals-16-01617-f006:**
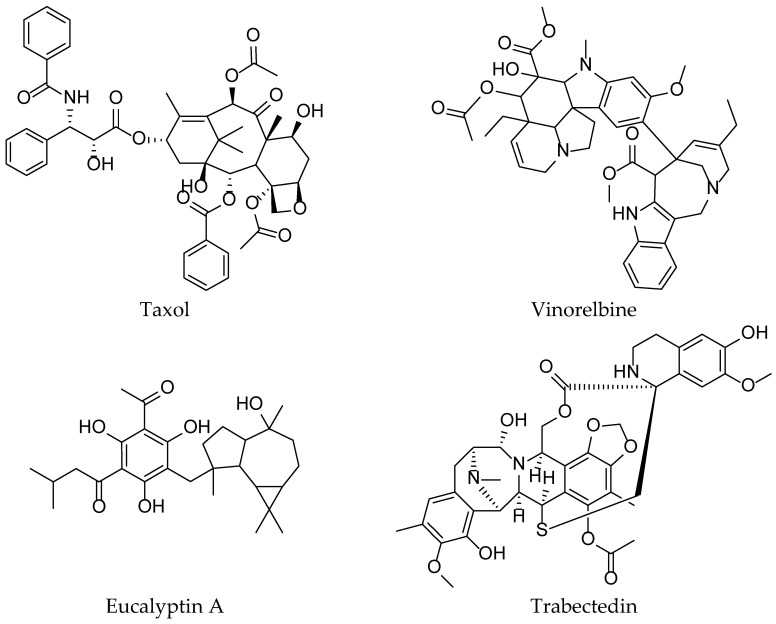
Natural molecule for cancer treatment.

**Figure 7 pharmaceuticals-16-01617-f007:**
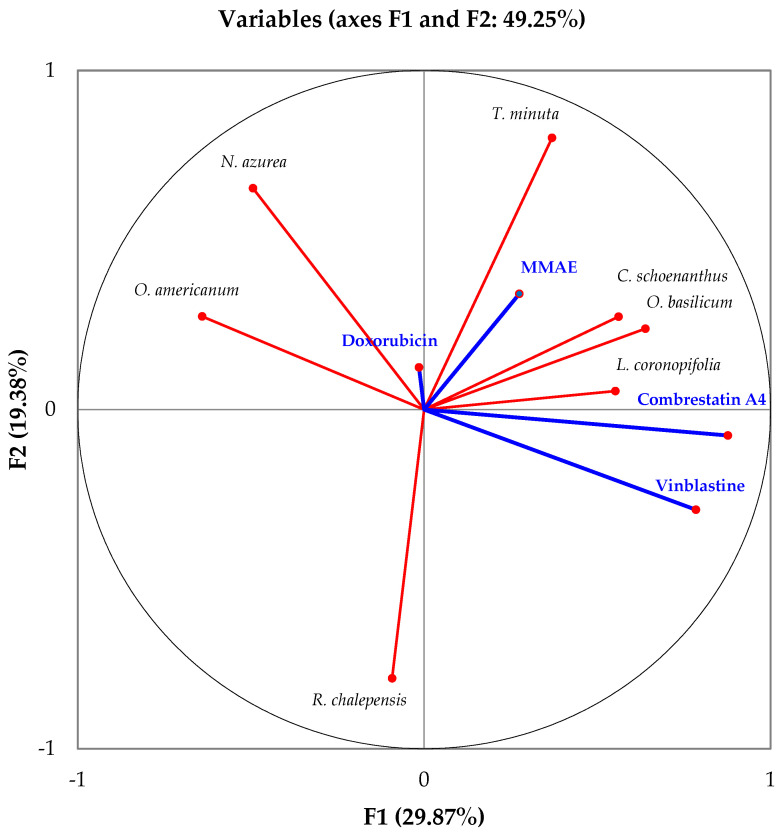
Correlations between the IC50 values of the products tested.

**Figure 8 pharmaceuticals-16-01617-f008:**
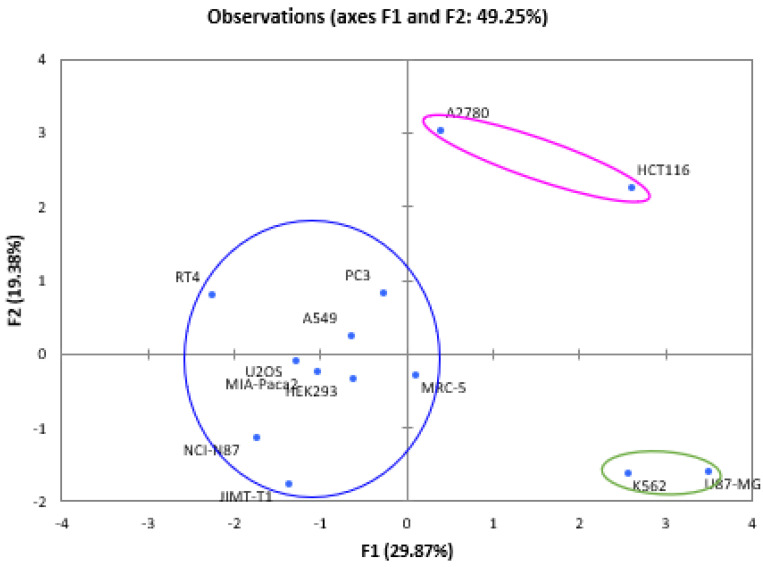
Biplot of the correlation between samples tested and cancer cell lines.

**Figure 9 pharmaceuticals-16-01617-f009:**
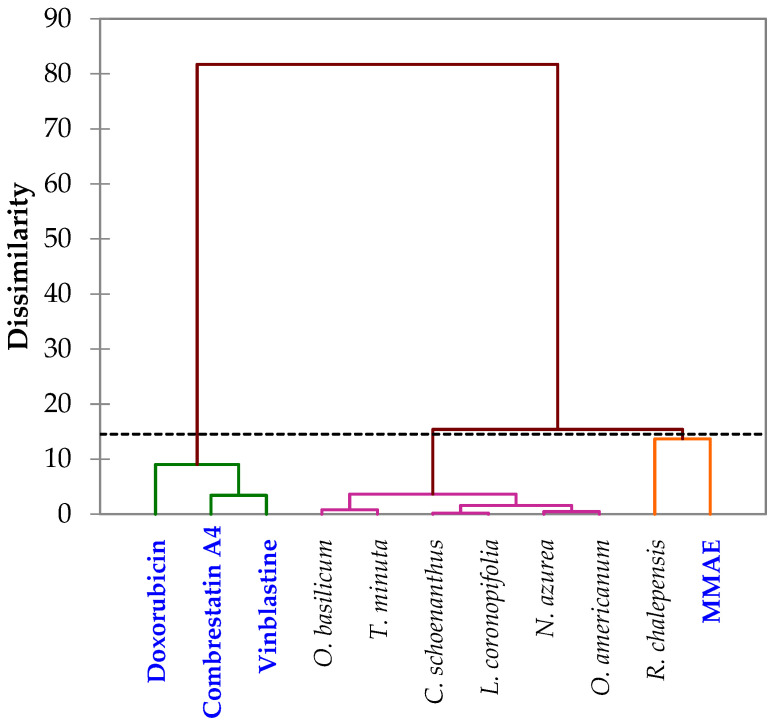
Cluster analysis of the studied and tested essential oils and drugs.

**Table 1 pharmaceuticals-16-01617-t001:** Numerical coding of the 50% inhibition concentration (*IC50*).

Concentration (µg/mL)	[0; 1]	[1; 5]	[5; 10]	[10; 20]	[20; 100]	>100
Code	5	4	3	2	1	0

**Table 2 pharmaceuticals-16-01617-t002:** Summary of 2020 cancer statistics in Djibouti as provided by the World Health Organization (WHO, 2021).

Statistics	Males	Females	Both Sexes
New cancer cases (‰)	0.29	0.48	0.77
Age-standardized incidence rate (world)	71.5	112.9	91.0
Risk of developing cancer before the age of 75 years (%)	7.9	11.7	9.7
Cancer deaths (‰)	0.22	0.32	0.54
Age-standardized mortality rate (world)	55.2	76.9	65.3
Risk of dying from cancer before the age of 75 years (%)	6.4	8.5	7.4
5-year prevalent cases (‰)	0.48	0.87	1.35
Top 5 most frequent cancers (*)	Prostate Colorectum Non-Hodgkin lymphoma Leukemia Liver	Breast Cervix uteri Ovary Colorectum Thyroid	Breast Cervix uteri Colorectum Non-Hodgkin lymphoma Leukemia

(*) Non-melanoma skin cancer is not included in the statistical analysis of the top five most common cancers.

**Table 3 pharmaceuticals-16-01617-t003:** Key components of comprehensive cancer control strategies.

**Prevention and Education**	Promoting awareness about modifiable risk factors such as tobacco use, unhealthy diet, and sedentary lifestyles plays a pivotal role. Public health campaigns, education initiatives, and policies targeting lifestyle changes are essential.
**Screening and Early Detection**	Implementing efficient screening programs for high-risk populations aids in identifying cancer at its nascent stage. Regular screenings for breast, cervical, colorectal, and other cancers enable timely interventions, significantly improving survival rates.
**Treatment Access and Quality Care**	Ensuring equitable access to quality cancer treatment is imperative. A robust healthcare infrastructure, skilled medical professionals, and up-to-date treatment protocols are critical to enhancing patient outcomes.
**Research and Innovation**	Continuous research fosters advancements in cancer prevention, diagnosis, and treatment. Investment in research infrastructure and clinical trials fuels progress in oncology.
**Policy of collaboration**	Effective policies facilitate cancer control efforts. Collaborations with governments, organizations, and stakeholders can drive policy changes that support cancer prevention and care.

**Table 4 pharmaceuticals-16-01617-t004:** Fundamental oncology services for effective care.

**Diagnosis and Staging**	Accurate diagnosis is paramount for designing personalized treatment plans. State-of-the-art diagnostic tools, including imaging techniques, biopsies, and molecular profiling, aid in precisely determining cancer type and stage.
**Multidisciplinary Treatment**	A multidisciplinary approach involving medical, surgical, and radiation oncologists, along with supportive care professionals, ensures a comprehensive treatment strategy tailored to the patient’s unique needs.
**Precision Medicine**	Molecular profiling guides treatment decisions by identifying genetic alterations driving cancer growth. Targeted therapies and immunotherapies maximize treatment efficacy while minimizing side effects.
**Pain and Symptom Management**	Palliative care focuses on alleviating pain and improving the quality of life for cancer patients. Comprehensive symptom management ensures comfort and enhances overall well-being.
**Psychosocial Support**	Cancer takes a toll on patients’ mental and emotional well-being. Access to counseling, support groups, and psychosocial services helps patients and their families cope with the challenges of diagnosis and treatment.

**Table 5 pharmaceuticals-16-01617-t005:** Place of collection of medicinal plants and their major chemical composition for essential oils.

Medicinal Plant	Place of Collection	Major Composition
*Cymbopogon schoenanthus* (L.) *Spreng*.	Mouloud	3-isopropenyl-5-methyl-1-cyclohexene (32.3%) D-limonene (11.3%)
*Lavandula coronopifolia* Poir.	Day	cis-caryophyllene (18.9%) Dehydronerolidol (12.8%) Isolongifolanone (11.2%)
*Nepeta azurea* R.Br	Day	Methyl (2E)-2-nonenoate (53.2%)
*Ocimum americanum* L.	Day	Carvotanacetol (38.4%) Estragole (27.5%)
*Ocimum basilicum* L.	Ambouli	Linalool (41.2%) Estragole (30.1%)
*Ruta chalepensis* subsp. *fumariifolia*	Day	2-undecanone (51.3%) Octyl acetate (17.3%)
*Tagetes minuta* L.	Day	Dihydrotagetone (20.8%) Artemisia (17.9%) (Z)-Tagetenone (12.4%) (-)-Spathulenol (11%)

**Table 6 pharmaceuticals-16-01617-t006:** The origin, source, and growth medium of 13 cancer cell lines used in the cytotoxicity test.

Cell Lines	Origin	Source
A2780	Ovarian carcinoma	ECACC-93112517
A549	Lung carcinoma	ATCC^®^-CCL-185TM
HCT116	Colorectal carcinoma	ATCC^®^-CCL-247TM
HEK-293	Embryonic kidney	ATCC^®^-CRL-1573TM
JIMT-T1	Breast carcinoma	DSMZ-ACC 589
K562	Myelogenous leukemia	ATCC^®^-CCL-243TM
MIA-Paca2	Pancreas carcinoma	ATCC^®^-CRL-1420TM
MRC5	Lung normal	ATCC^®^-CCL-171TM
NCI-N87	Gastric carcinoma	ATCC^®^-CRL-5822TM
PC3	Prostate carcinoma	ATCC^®^-CRL-1435TM
RT4	Urinary bladder	ATCC^®^-HTB-2TM
U2OS	Bone osteosarcoma	ATCC^®^-HTB-96TM
U87-MG	Brain glioblastoma	ATCC^®^-HTB-14TM

**Table 7 pharmaceuticals-16-01617-t007:** Results for cytotoxicity activity of Djibouti essential oils.

Cell Line	*C. schoenanthus*	*L. coronopifolia*	*N. azurea*	*O. americanum*	*O. basilicum*	*R. chalepensis*	*T. minuta*	Combrestatin A4	Doxorubicin	MMAE	Vinblastine
A2780	0.14 ± 0.03	0.21 ± 0.01	0.62 ± 0.09	0.69 ± 0.02	1.01 ± 0.11	>100	0.36 ± 0.05	-	-	0.45 ± 0.01	-
A549	0.49 ± 0.23	0.92 ± 0.14	0.07 ± 0.01	0.87 ± 0.06	5.37 ± 0.16	8.22 ± 0.73	1.57 ± 0.73	20.00 ± 0.10	56.60 ± 0.84	0.46 ± 0.05	-
HCT116	0.65 ± 0.03	0.25 ± 0.03	0.11 ± 0.01	1.01 ± 0.01	1.77 ± 0.07	>100	0.47 ± 0.04	2.00 ± 0.10	-	2.07 ± 0.02	35.00 ± 0.84
HEK293	0.19 ± 0.05	0.12 ± 0.05	0.83 ± 0.11	0.25 ± 0.03	1.40 ± 0.11	1.39 ± 0.27	1.20 ± 0.32	-	-	-	-
JIMT-T1	1.50 ± 0.30	0.71 ± 0.03	2.07 ± 0.20	0.92 ± 0.03	5.46 ± 0.051	6.66 ± 0.15	2.13 ± 0.38	-	-	-	-
K562	0.99 ± 0.01	0.67 ± 0.15	1.00 ± 0.01	1.01 ± 0.01	3.67 ± 0.65	3.08 ± 0.59	1.06 ± 0.05	5.00 ± 0.30	-	3.12 ± 0.2	20.00 ± 0.12
MIA-Paca2	0.55 ± 0.02	0.45 ± 0.06	0.86 ± 0.01	0.99 ± 0.03	5.31 ± 0.17	4.84 ± 0.04	1.61 ± 0.06	-	-	4.36 ± 0.2	-
MRC-5	0.83 ± 0.09	0.12 ± 0.01	0.09 ± 0.01	1.34 ± 0.16	5.48 ± 0.182	7.85 ± 0.13	1.33 ± 0.14	-	39.88 ± 1.22	-	-
NCI-N87	3.26 ± 1.52	4.22 ± 1.38	0.90 ± 0.27	4.28 ± 0.83	3.43 ± 0.12	2.31 ± 1.28	1.48 ± 0.10	-	-	1.65 ± 0.07	-
PC3	0.29 ± 0.01	0.97 ± 0.07	0.80 ± 0.07	0.95 ± 0.02	4.37 ± 1.02	8.97 ± 0.17	1.71 ± 0.18		2.09 ± 0.03	0.36 ± 0.03	-
RT4	4.75 ± 1.24	1.57 ± 0.73	0.53 ± 0.01	0.86 ± 0.02	6.89 ± 1.52	>100	1.37 ± 0.29	-	36.29 ± 1.20	0.50 ± 0.01	-
U2OS	0.24 ± 0.02	1.28 ± 0.14	0.67± 0.05	0.68 ± 0.05	4.29 ± 0.97	5.45 ± 0.76	1.16 ± 0.04	-	-	-	-
U87-MG	0.59 ± 0.09	0.34 ± 0.04	1.13 ± 0.22	1.31 ± 0.62	4.29 ± 0.65	6.03 ± 0.49	1.01 ± 0.12	9.00 ± 0.50	99.61 ± 2.34	0.21 ± 0.03	2.00 ± 0.04

**Table 8 pharmaceuticals-16-01617-t008:** New parameters for numerical coding of the efficacy of the products tested against cancer cell lines.

Cell Line	*C. schoenanthus*	*L. coronopifolia*	*N. azurea*	*O. americanum*	*O. basilicum*	*R. chalepensis*	*T. minuta*	Combrestatin A4	Doxorubicin	MMAE	Vinblastine
A2780	5	5	5	5	4	0	5	0	0	5	0
A549	5	5	5	5	3	3	4	1	1	5	0
HCT116	5	5	5	4	4	0	5	4	0	4	1
HEK293	5	5	5	5	4	4	4	0	0	0	0
JIMT-T1	4	5	4	5	3	3	4	0	0	0	0
K562	5	5	4	4	4	4	4	4	0	4	1
MIA-Paca2	5	5	5	5	3	4	4	0	0	4	0
MRC-5	5	5	5	4	4	3	4	0	1	0	0
NCI-N87	4	4	5	4	3	4	4	0	0	4	0
PC3	5	5	5	5	4	3	4	0	4	5	0
RT4	4	4	5	5	3	0	4	0	1	5	0
U2OS	5	4	5	5	4	3	4	0	0	0	0
U87-MG	5	5	4	4	4	3	4	3	1	5	4

## Data Availability

Data is contained within the report.

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
