# Peer review of "An Overview of Cancer in Djibouti: Current Status, Therapeutic Approaches, and Promising Endeavors in Local Essential Oil Treatment"

_pharmaceuticals, 2023, doi:10.3390/ph16111617_

Round 1

Reviewer 1 Report

Comments and Suggestions for Authors

The current review debates the Cancer in Djibouti and a few factors that may interfere and may be related to it. Here goes a few concerns.

The corresponding author contacts are missing.

I recommend the authors to place the keywords by alphabetic order.

Personal pronouns should be avoided in both Abstract and Manuscript Body.

A proper aim for the present review is missing in the end of the Introduction section.

In Figure 1 the coordinates are not needed. Only the name is fine.

Before the “Conclusions” I would recommend the authors to add a subheading regarding “Limitations of the current knowledge in Djibouti” and another “Further knowledge perspectives”.

Additionally, the “Conclusions” section I would rename it to “Final remarks”, since this is not a research study but a review instead.

Globally it make a good overview.

Author Response

Dear Reviewer,

I wish to extend my team's heartfelt appreciation for the insightful comments and suggestions you provided for our work, which have undoubtedly enriched the entire manuscript.

Below, you will find our responses to your feedback.

Thank you in advance for your kind consideration. Please accept, Sir, the assurance of our utmost respect.

Reviewer 1 :

The current review debates the Cancer in Djibouti and a few factors that may interfere and may be related to it. Here goes a few concerns.

1) The corresponding author contacts are missing.

We have added the email of the corresponding author

2) I recommend the authors to place the keywords by alphabetic order.

we have placed the keywords in alphabetical order

3) Personal pronouns should be avoided in both Abstract and Manuscript Body.

 we have made as much correction as possible to minimize personal pronouns

4) A proper aim for the present review is missing in the end of the Introduction section.

We've added a detailed objective at the end of the Introduction section.

5) In Figure 1 the coordinates are not needed. Only the name is fine.

 We removed the coordinates in Figure 1

6) Before the “Conclusions” I would recommend the authors to add a subheading regarding “Limitations of the current knowledge in Djibouti” and another “Further knowledge perspectives”.

We have added paragraph 3.9 which concerns “The limits of current knowledge in Djibouti - perspectives”.

7) Additionally, the “Conclusions” section I would rename it to “Final remarks”, since this is not a research study but a review instead.

we have made the requested corrections

Reviewer 2 Report

Comments and Suggestions for Authors

This research reported the cancer in Djibouti, as a project report, this paper clearly shows the current status, the cancer risk factors, cancer control, and medical treatment for cancers. This is a relatively new analytical review article. However, there are several questions that should be addressed before publications.

1There is a certain disconnect between the first part of the article and the content of the treatments that follow, and in the middle it is suggested to include treatments for specific cancers, such as which types of cancers are more common and where there are fewer cancers

2The latter part reads like a review of cancer treatments and whether it is possible to target table complaints for cancers in Djibouti.

3Excellent drug loading and controlled release are important in the treatment of anticancer, and the authors are advised to explore this point.

4Some of the advantages of drug delivery by polymer should be cited and compare, such as Biomacromolecules 2021, 22 (2) , 732-742; Pharmaceutics 2023, 15(2), 368

Author Response

Dear Reviewer,

Please find enclosed the responses to the requested comments and remarks.

We thank you for the interest you have shown in our manuscript.

Looking forward to a favorable response, please accept our kind regards.

Best regards,

1:There is a certain disconnect between the first part of the article and the content of the treatments that follow, and in the middle it is suggested to include treatments for specific cancers, such as which types of cancers are more common and where there are fewer cancers

The disparity between the initial part of the article and the subsequent treatment content may stem from the initial aim of providing a comprehensive overview before delving into specific cases. We have incorporated detailed information on the most common types of cancers and their geographical prevalence, along with preventive measures. (Table 2 and Table 6)

2:The latter part reads like a review of cancer treatments and whether it is possible to target table complaints for cancers in Djibouti.

In response to the feedback, we have included an additional section in the final part, emphasizing the challenges encountered and the potential prospects for addressing cancer-related issues, particularly within the context of Djibouti.

3:Excellent drug loading and controlled release are important in the treatment of anticancer, and the authors are advised to explore this point.

Indeed, optimal drug loading and controlled release are critical elements in the treatment of anticancer drugs. These aspects are particularly important for several key reasons:

  1. Increased Therapeutic Efficacy: Proper drug loading allows for the precise administration of the required amount of medication to eradicate cancer cells while minimizing undesirable side effects. Controlled release ensures that the drug is diffused in a controlled manner in the body, helping to maintain optimal therapeutic concentrations over a specific period.
  2. Reduction of Side Effects: By controlling the drug release, peaks in drug concentration in the blood can be avoided, thereby reducing the harmful side effects associated with many anticancer treatments. This also helps improve patient tolerance to the treatment.
  3. Enhanced Treatment Compliance: Controlled release can also reduce the frequency of drug administrations, improving treatment compliance, especially among patients who struggle with complex or frequent treatment regimens.
  4. Increased Drug Bioavailability: By optimizing drug loading, the drug's bioavailability in the body can be improved, enhancing its therapeutic efficacy and reducing the likelihood of developing resistance.
  5. Improved Pharmacokinetics: Controlled release can help maintain stable therapeutic concentrations over an extended period, which is particularly crucial for anticancer drugs with a narrow therapeutic margin.

To achieve these goals, researchers are exploring a variety of strategies, such as the use of nanotechnologies, specific polymers, targeted carriers, and other advanced drug delivery systems, to optimize the loading and release of anticancer drugs. This approach improves the overall effectiveness of the treatment while reducing undesirable side effects.

Reviewer 3 Report

Comments and Suggestions for Authors

The paper “An Overview of Cancer in Djibouti: Current Status, Therapeutic Approaches, and Promising Endeavors in Local Essential Oils Treatment‘’

This comprehensive overview covers a wide spectrum, starting with the current state of affairs-an investigation into cancer risk factors. 

The paper is prepared professionally. It includes a well-crafted abstract and an exhaustive introduction that justifies the research undertaken. The introduction points to the deficiencies in the literature on the subject. The aim is clearly defined. Modern analytical methods were used in the research. The discussion of the results is well prepared. The conclusions are well-defined. The illustrative material is appropriate.

In my opinion, the manuscript after corrections, will be suitable for publication in a journal.

Detailed comments:

Abstract: Should include some numeric data obtained from the study (if any)

Do not use abbreviations when use first time.

Introduction - The introduction is enough in my opinion. Introduction needs some minor changes

Line 55-57 Medicinal plants 56 rich in antioxidants safeguard cells from harm, yielding preventative benefits against cancer 57 and other diseases [14]. Please add more references. I suggest below ones.

KHAMMASSI, MARWA; AYED, RAYDA BEN; KHEDHIRI, SANA; SOUIHI, MOUNA; HANANA, MOHSEN; AMRI, ISMAIL; and HAMROUNI, LAMIA.  "Crude extracts and essential oil of Platycladus orientalis (L.) Franco: a source of phenolics with antioxidant and antibacterial potential as assessed through a chemometric approach. Turk J Agric For 2022, 46 (4): 477-487. https://doi.org/10.55730/1300-011X.3019.

ÇELEBİ, ÖZGÜR; FIDAN, HAFIZE; ILIEV, IVAN; PETKOVA, NADEZHDA; DINCHEVA, IVAYLA; GANDOVA, VANYA; STANKOV, STANKO; and STOYANOVA, ALBENA (2023) "Chemical composition, biological activities, and surface tension properties of Melissa officinalis L. essential oil," Turkish Journal of Agriculture and Forestry: 47 (1): 67-78. https://doi.org/10.55730/1300-011X.3065.

Table 1. (IC50). IC should be italic

Figure 2 Gentics must be genetics and any references for Figure 2 I mean any copyright?????

Cancer risk factors section....Each parameters tobacco etc. needs at least 3-4 clinical confirmed result papers as reference

Comments on the Quality of English Language

na

Author Response

Dear Reviewer,

First and foremost, on behalf of my team, I would like to express our gratitude for the comments and feedback you provided for our work, which have significantly enhanced the manuscript.

Below, you will find the responses to your comments.

Thank you in advance for your consideration. Please accept, Sir, the assurance of our highest regards.

Reviewer 3 :

1) Abstract: Should include some numeric data obtained from the study (if any)

The digital data in this overview is limited; we have merely rephrased the summary.

2)Do not use abbreviations when use first time.

We have taken this comment into account throughout the entire manuscript.

3) Introduction - The introduction is enough in my opinion. Introduction needs some minor changes

We have rephrased the final paragraph and clarified the overall aim of the study.

4) Line 55-57 Medicinal plants 56 rich in antioxidants safeguard cells from harm, yielding preventative benefits against cancer 57 and other diseases [14]. Please add more references. I suggest below ones.

KHAMMASSI, MARWA; AYED, RAYDA BEN; KHEDHIRI, SANA; SOUIHI, MOUNA; HANANA, MOHSEN; AMRI, ISMAIL; and HAMROUNI, LAMIA.  "Crude extracts and essential oil of Platycladus orientalis (L.) Franco: a source of phenolics with antioxidant and antibacterial potential as assessed through a chemometric approach. Turk J Agric For 2022, 46 (4): 477-487. https://doi.org/10.55730/1300-011X.3019.

ÇELEBİ, ÖZGÜR; FIDAN, HAFIZE; ILIEV, IVAN; PETKOVA, NADEZHDA; DINCHEVA, IVAYLA; GANDOVA, VANYA; STANKOV, STANKO; and STOYANOVA, ALBENA (2023) "Chemical composition, biological activities, and surface tension properties of Melissa officinalis L. essential oil," Turkish Journal of Agriculture and Forestry: 47 (1): 67-78. https://doi.org/10.55730/1300-011X.3065.

We have added the requested references to provide a scientific perspective on the mechanisms behind the antioxidant activity of medicinal plants for cancer prevention.

5) Table 1. (IC50). IC should be italic

We have made the requested correction.

6) Figure 2 Gentics must be genetics and any references for Figure 2 I mean any copyright?????

Regarding Figure 2, which pertains to the main cancer risk factors, the displayed images were created by us using Edraw software.

7) Cancer risk factors section....Each parameters tobacco etc. needs at least 3-4 clinical confirmed result papers as reference

We have added relevant references that pertain to this section.

Round 2

Reviewer 1 Report

Comments and Suggestions for Authors

Dear authors, I have no further concern.